# Carotid Stenosis Assessment with Vector Concentration before and after Stenting

**DOI:** 10.3390/diagnostics10060420

**Published:** 2020-06-20

**Authors:** Andreas Hjelm Brandt, Tin-Quoc Nguyen, Henrik Gutte, Jonathan Frederik Carlsen, Ramin Moshavegh, Jørgen Arendt Jensen, Michael Bachmann Nielsen, Kristoffer Lindskov Hansen

**Affiliations:** 1Department of Radiology, Copenhagen University Hospital, Rigshospitalet, 2100 Copenhagen, Denmark; tin-quoc.nguyen.02@regionh.dk (T.-Q.N.); henrik.gutte.borgwardt.04@regionh.dk (H.G.); jonathan.frederik.carlsen@regionh.dk (J.F.C.); mbn@dadlnet.dk (M.B.N.); lindskov@gmail.com (K.L.H.); 2Department of Clinical Medicine, University of Copenhagen, 2200 Copenhagen, Denmark; 3BK Medical, 2730 Herlev, Denmark; RMoshavegh@bkmedical.com; 4Center for Fast Ultrasound Imaging, Department of Health Technology, Technical University of Denmark, 2800 Lyngby, Denmark; jaj@elektro.dtu.dk

**Keywords:** vector concentration, vector flow imaging, duplex ultrasound, digital subtraction angiography, stenosis degree, carotid artery stenosis

## Abstract

Digital subtraction angiography (DSA) is considered the reference method for the assessment of carotid artery stenosis; however, the procedure is invasive and accompanied by ionizing radiation. Velocity estimation with duplex ultrasound (DUS) is widely used for carotid artery stenosis assessment since no radiation or intravenous contrast is required; however, the method is angle-dependent. Vector concentration (VC) is a parameter for flow complexity assessment derived from the angle independent ultrasound method vector flow imaging (VFI), and VC has shown to correlate strongly with stenosis degree. The aim of this study was to compare VC estimates and DUS estimated peak-systolic (PSV) and end-diastolic velocities (EDV) for carotid artery stenosis patients, with the stenosis degree obtained with DSA. Eleven patients with symptomatic carotid artery stenosis were examined with DUS, VFI, and DSA before and after stent treatment. Compared to DSA, VC showed a strong correlation (*r* = −0.79, *p* < 0.001), while PSV (*r* = 0.68, *p* = 0.002) and EDV (*r* = 0.51, *p* = 0.048) obtained with DUS showed a moderate correlation. VFI using VC calculations may be a useful ultrasound method for carotid artery stenosis and stent patency assessment.

## 1. Introduction

The presence of carotid artery stenosis is clinically relevant, as 10–15% of all ischemic strokes are caused by carotid lumen reduction [1]. A precise diagnosis of carotid artery stenosis is crucial in the decision of the optimal treatment for each patient [2]. Carotid artery stenting (CAS) is an endovascular procedure for the treatment of carotid artery stenosis, where a stent is inserted in the lumen, reducing the risk of recurrent stroke significantly, and is often used when the surgical procedure with plaque removal carotid endarterectomy (CEA) is not suitable [3].

Digital subtraction angiography (DSA) is considered the reference method for carotid artery stenosis assessment; however, the procedure is invasive and accompanied by ionizing radiation for the patient [4,5]. Duplex ultrasound (DUS) using spectral Doppler ultrasound is also widely used for carotid artery stenosis assessment and for treatment decision-making, as it is non-ionizing and does not require the administration of an intravenous contrast agent. Furthermore, DUS is less expensive and more accessible than DSA and other imaging modalities such as computed tomography angiography (CTA) and magnetic resonance angiography. DUS estimates the severity of the stenosis by determining peak systolic velocity (PSV) and end-diastolic velocity (EDV) [6]. The main drawback of DUS is angle dependency, as only the velocity along the axis of the emitted beam is estimated. A valid velocity estimate, therefore, necessitates manual angle correction [7]. In complex flow conditions, the manual angle correction is prone to errors, which probably is the main reason for DUS being greatly susceptible to operator-dependent variation [6,8,9].

An alternative to DUS is the angle independent ultrasound technique Vector Flow Imaging (VFI) [10]. VFI has fewer operator-dependent settings and a superior intra- and interobserver agreement compared to DUS [11,12,13]. VFI measures both the axial and the transverse velocity components of the blood flow in real-time, thereby permitting vector velocity estimation [14,15]. The complexity of the flow can be quantified as vector concentration (VC), a VFI derived parameter for flow assessment [16]. VC has previously been investigated for complex flow assessment in patients with aortic valve and femoral artery stenosis, indicating high agreement with stenosis degree [17,18,19,20], and examples of VC assessment of carotid artery stenosis with promising results have also been published [21,22].

In this preliminary study, patients with symptomatic carotid artery stenosis were examined with VFI, DUS, and DSA. The study aimed to compare VC obtained with VFI and DUS velocities with the obtained DSA stenosis degree before and after CAS treatment in carotid artery stenosis patients. The hypothesis was that VFI can detect significant carotid stenosis and assess the effect of stenting similar to DUS when compared with DSA.

## 2. Materials and Methods

Eleven patients (8 males, 3 females, mean age: 64.5 years, range: 47–81 years) with proximal internal carotid artery (ICA) or common carotid artery (CCA) stenosis were included. All patients presented with a carotid artery stenosis degree of >60%, determined beforehand with CTA. All patients had symptomatic carotid artery stenosis with ipsilateral neurological symptoms and were treated with CAS within a week after onset of symptoms. Only patients examined with DSA, DUS, and VFI before and after CAS treatment were included in the study analysis. Due to human error, DUS data were not obtained for one patient before CAS treatment and this patient was excluded. The final study population is presented in Table 1. The local ethics committee waived ethical approval as ultrasound scanning of the carotid is considered a routine procedure (no.: H-19009278, 5 March 2019. This study was approved by the Danish Data Protection Agency (Approval: VD-2019-238, 10 May 2019)).

All patients were scanned with a commercial ultrasound scanner (BK5000, BK Medical, Herlev, Denmark) equipped with a linear transducer using a frequency range of 2–8 MHz (8L2, BK Medical, Herlev, Denmark). The scanner was equipped with a commercially available setup for VFI and DUS. Both VFI and DUS examinations were performed in the angio-suite just before and after DSA and CAS treatment. The ultrasound scan included an examination of both the prestenotic- and stenotic regions with VFI and DUS.

In the VFI scan setup, the vessel was scanned with a long-axis view. If turbulent/disturbed flow was detected, the area was inspected for aliasing, indicating increased blood flow velocity. The pulse repetition frequency (PRF) was set to the lowest setting, where no aliasing was visible on the scanner display and with optimal filling of the vessel at the same time. Video sequences were obtained with the vessel centered in the scan plane. The flow through the stenosis was evaluated with VC as a mean over 300 frames, with a frame rate of 30 Hz. To avoid blooming artifacts, the wall filter and color gain were adjusted for optimal filling of the vessel, while all other settings stayed in default mode. The angle of insonation was 70–90 degrees in all cases, and the maximum scan depth was approximately 3 cm. VFI recordings were processed offline in MATLAB (MathWorks, Natick, MA, USA), as previously described [16,23].

VFI estimates both the axial and transverse velocity components using emissions of conventional pulses for Doppler ultrasound. The vectors in the axial direction are found similar to conventional Doppler ultrasound, while the motion in the transverse direction is found by using a changed apodization in receive beamforming and a special estimator [14,15]. The vector flow is displayed within a color box with the direction and magnitude shown by a color wheel. VC estimates the vector angle distribution within a manually marked region of interest (ROI) (Figure 1). In complex flow, the VC goes towards 0, while VC goes towards 1 for laminar flow. The ROI adjusted to the diseased vessel segment before stenting was placed in the same vessel segment after stenting for comparison. The average VC of the blood flow in the diseased vessel segment was found during approximately 5 consecutive systoles, with a minimum of 150 frames. Previous papers provide detailed explanations of the VC calculation [16,17].

DUS data were obtained with the same ultrasound scanner and transducer as for VFI data. A commercially available standard setup for spectral Doppler was used for velocity estimation without any post-processing. In the DUS scan setup, the vessel was scanned in a long-axis view, and the PRF was set to the lowest possible setting without aliasing. The range gate was placed in the mid lumen of the stenotic area, and manual angle correction was performed according to the assumed flow direction. Angle correction was 60 degrees or below for all scans. Spectral Doppler curves with PSV and EDV were documented for each scan (Figure 2).

A biplane angiography system (Artis Q, Siemens Healthcare GmbH, Erlangen, Germany) was used to perform the DSA for CAS treatment guidance, and the procedure was performed either in local or general anesthesia. All patients were on dual platelet treatment prior to the procedure. A 6- or 9-Fr sheath was placed in the femoral artery. A 4- or 5-Fr catheter was used when needed for contrast injections at the place of interest. DSA of the carotid artery was performed using 2 frames/s and a 6–10 mL iodine contrast injection (Visipaque 270 mgI/mL, GE Healthcare, Chicago, Illinois, USA). Stent treatments were performed by four different neurointerventionalists and all patients were treated with a Casper carotid stent (MicroVention Inc., Aliso Viejo, CA, USA), covering the stenotic area using only one stent. No technical failures were detected during the produces, and there were no complications. The DSA image, which was obtained in two planes yielding the most severe diameter reduction, was used for calculation of the stenosis degree for each patient (Figure 3). The stenosis degree was calculated according to The European Carotid Surgery Trial (ECST) method [24].

Measurements obtained with VFI, DUS, and DSA were analyzed with descriptive statistics, i.e., mean, standard deviation (SD), and confidence interval. Measurements before and after stent treatment were analyzed with a paired 2-tailed *t*-test. Spearman’s Rank Correlation was used to test the linear relationship between VC, PSV, and EDV against the DSA stenosis degree. A *p*-value < 0.05 was considered significant. Statistical analyses and data handling were performed with IBM SPSS Statistics (SPSS Inc., Chicago, IL, USA) and Microsoft Excel (Microsoft Corporation, Redmond, WA, USA).

## 3. Results

Mean values and mean differences between VC, PSV, EDV, and mean stenosis degree are given in Table 2. Changes in VC, PSV, and stenosis degree but not EDV were significantly different after CAS treatment (Table 2).

The stenosis degree obtained with DSA had a strong inverse correlation to VC (*r* = −0.79, *p* < 0.001), while correlations to PSV (*r* = 0.68, *p* = 0.002) and EDV (*r* = 0.51, *p* = 0.048) were moderate. Scatterplots of comparisons are given in Figure 4.

## 4. Discussion

Stenosis degree obtained with DSA, VC obtained with VFI and PSV with DUS were able to assess the carotid artery stenosis and treatment response after CAS. In addition, VC correlated strongly with stenosis degree in the carotid artery, while PSV and EDV only correlated moderately. This indicates that VC is able to detect significant carotid stenosis and assess the effect of stenting similar to PSV estimated with DUS. Thus, flow assessment with VC may be an aid or even a better estimate for the noninvasive assessment of carotid artery stenosis and stent patency, compared to DUS. This study supports results of previous studies with VC, where VC assessment of femoral artery stenosis had a strong linear correlation with the DSA stenosis degree (*r* = 0.93, *p* < 0.001), and of aortic valve stenosis, where transesophageal echocardiography derived PSV was significantly associated with VC (*r* = 0.88, *p* < 0.0001) [17,18].

DUS is used widely to detect carotid artery stenosis and to select patients for CAS or CEA, however, sensitivity and specificity vary widely between studies. A sensitivity ranging from 76% to 90% and specificity from 75% to 94% have been reported for a >70% stenosis degree [25,26,27]. Utilizing a method with a higher diagnostic accuracy could potentially be an aid in the decision of using either CAS or CEA treatment for patients with carotid artery stenosis. VFI provides more flow data compared to DUS, as flow direction and velocity within every pixel of an ROI are given. Furthermore, all flow data within the ROI are used for VC calculations i.e., more data are used for the flow evaluation when compared to conventional velocity estimation using DUS.

Flow complexity can be assessed with spectral Doppler by estimating spectral broadening or by evaluating mosaic patterns in color Doppler; however, none of these methods are quantitative like VC is [28,29]. Moreover, VC seems to be a more powerful tool for complex flow pattern recognition than DUS in the ICA [30]. Quantifying flow complexity with VC was initially introduced to separate laminar and complex flow conditions in the normal carotid bulb [16]; however, flow complexity is also increased in vessels with pathology [20]. Previous VFI studies of carotid artery stenosis have indicated that flow complexity increases near and within a stenotic area [21,22].

Precise velocity estimation with DUS relies on correct manual angle correction; however, flow near a stenosis is seldom laminar, but rather complex, meaning that the peak velocities are not always found in the center of the vessel [31]. Additionally, the carotid artery is often parallel to the skin, resulting in velocity estimates obtained at beam-to-flow angles above 60 degrees. The manual adjustment of the Doppler angle can be incorrect and result in inaccurate velocity estimates [32,33]. Since VC calculations are based on VFI estimates, concerns about beam-to-flow angle and manual angle correction are obsolete. Employing VFI, for carotid stenosis assessment will, thus, overcome the shortcomings of velocity estimation with DUS [12,34]. Moreover, DUS estimates are highly dependent on the operator’s experience level [6]. VFI seems to be less affected by the operator’s level of experience compared to DUS, which may the same for VC [13].

There are several additional advantages with flow evaluation using VC. The presence of calcified plaques is common for carotid artery stenosis patients and results in acoustic shadowing. This obscures the DUS’s ability to obtain valid velocity estimates with a reported low sensitivity ranging from 22.7% to 32.5% [35]. Since all flow data within the ROI are used for flow estimation, VC is not dependent on flow alignment meaning that the VC estimate is less affected by acoustic shadowing [36]. Additionally, DUS is limited by aliasing, which is governed by the PRF setting. Large errors are present for the PSV estimate when aliasing occurs and flow is going in the wrong direction. Conversely, VC estimates have previously been indicated to be less PRF-sensitive, as indicated by several studies [17,18,19,37].

Several limitations should be acknowledged in this study. The most obvious limitation of this preliminary study is the small study population. Larger studies, preferably with multiple centers, are warranted to confirm the results. Additionally, only patients with a stenosis degree >60% and open stents were assessed, thus, the correlation analyses were based on two extreme data sets as seen in Figure 4, and therefore, a study concerning non-selected patients with non-significant stenosis is warranted. Alignment with DSA can be a problem, since mean PSV can vary 9 cm/s for each centimeter of distance from the bifurcation [38]. An incorrect alignment between the methods, before and after stenting could probably affect the outcome. Additionally, calculations of VC were not performed directly on the scanner but processed off-line by one single observer. An integrated fully automatic method for quantitative flow assessment could eliminate this disadvantage and ease the pursuit for a larger more evidence compiling study. Finally, as only one operator performed the examinations in this study, no interobserver variability was assessed.

## 5. Conclusions

This is the first preliminary study showing that VFI may be used to evaluate carotid artery stenosis and stent treatment with an improved correlation to DSA, when compared with the conventional DUS technique. The results indicate that VC is a useful parameter for stenosis assessment in the carotid artery, and that VC may be a parameter that should be added to the regular examination of patients suffering from carotid artery stenosis.

## Figures and Tables

**Figure 1 diagnostics-10-00420-f001:**
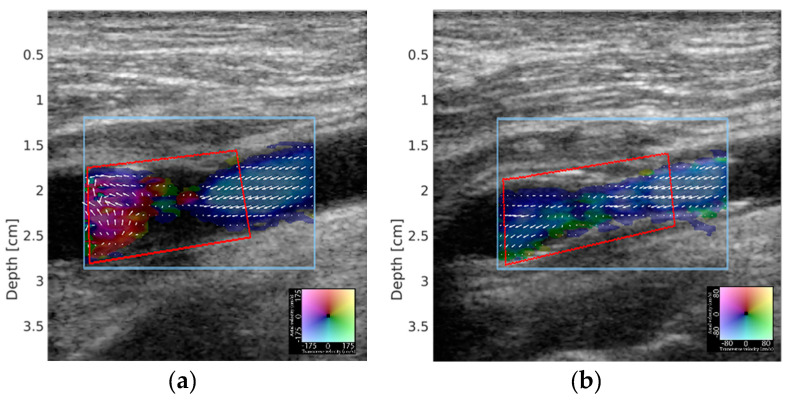
Vector flow imaging (VFI) example of a carotid artery stenosis patient (no. 10) before and after carotid artery stenting (CAS) treatment. In (**a**), complex flow is observed within the stenosis, while flow is almost laminar after stent treatment in (**b**). Vector concentration (VC) of flow in the stenosis was 0.45 before and 0.85 after stent treatment. VC was calculated within the ROI delineated by the red box.

**Figure 2 diagnostics-10-00420-f002:**
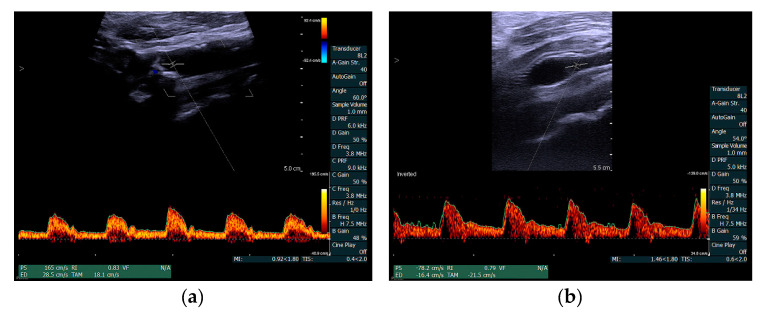
Duplex ultrasound (DUS) example of a carotid artery stenosis patient (no. 10) before and after stent treatment. Image (**a**) shows the DUS estimation of the stenotic site with peak-systolic velocity (PSV) of 165 cm/s and end-diastolic velocity (EDV) of 28.5 cm/s; Image (**b**) shows the DUS estimation after stent treatment with PSV reduced to 78.2 cm/s and EDV to 16.4 cm/s.

**Figure 3 diagnostics-10-00420-f003:**
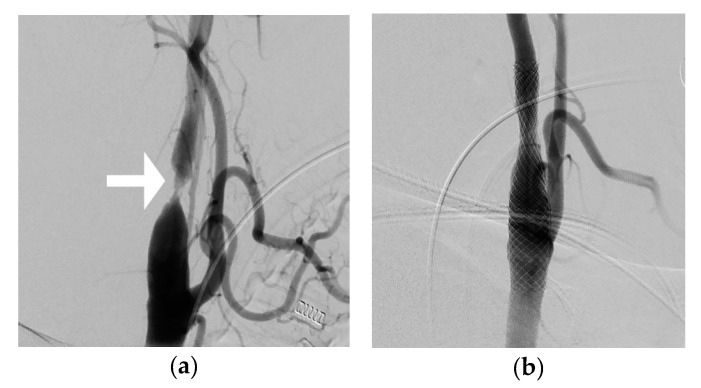
Digital subtraction angiography (DSA) example of a carotid artery stenosis patient (no. 10) before and after CAS treatment. Image (**a**) shows the stenotic part of the internal carotid artery (ICA) marked with a white arrow, while image (**b**) shows the same segment with a slightly different image projection after CAS treatment.

**Figure 4 diagnostics-10-00420-f004:**
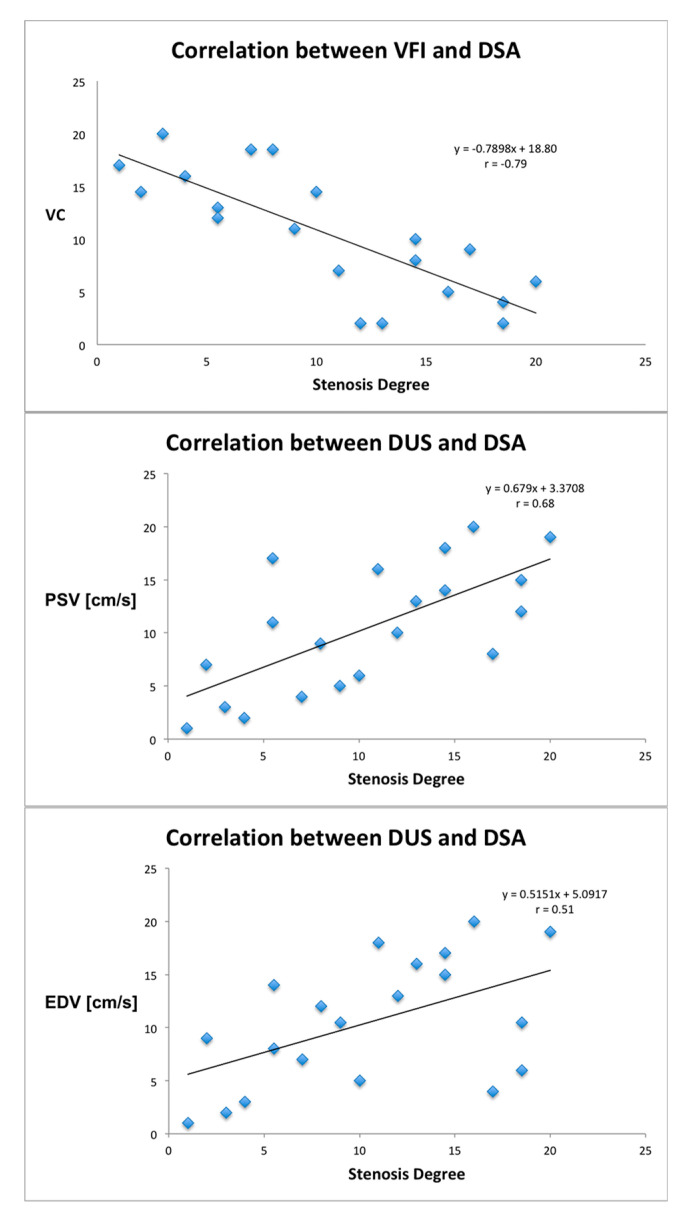
Scatterplots (spearman rank correlation) of VFI and DUS against DSA. The solid line in each plot indicates the line of best fit.

**Table 1 diagnostics-10-00420-t001:** Overview of stenosis site, sex, and age for the patients (n = 10) included in the study analysis.

Patient Number	Stenosis Site	Sex	Age (Year)
1	Right ICA	Male	66
2	Left CCA	Male	81
3	Left ICA	Female	68
4	Right CCA	Male	47
5	Left CCA	Female	52
6	Right ICA	Male	58
7	Right ICA	Female	71
8	Right ICA	Male	67
9	Left ICA	Male	48
10	Left ICA	Male	77

**Table 2 diagnostics-10-00420-t002:** Mean values and mean differences between VC, PSV, EDV, and mean stenosis severity.

	Before Stenting (SD)	After Stenting (SD)	Mean Difference [CI95]	*p*-Value
Mean VC	0.56 (10.7)	0.92 (5.3)	+0.39 [0.32; 0.46]	<0.001
Mean PSV (cm/s)	149.9 (79.6)	68.9 (20.5)	−81.2 [−104.38; −58.36]	0.005
Mean EDV (cm/s)	58.0 (69.8)	23.9 (13.2)	−34.1 [−70.8; 2.52]	0.102
Mean stenosis degree (%)	72.4 (8.2)	13.3 (7.0)	−59% [−52%; −67%]	<0.001

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
