# Peer review of "Carotid Stenosis Assessment with Vector Concentration before and after Stenting"

_diagnostics, 2020, doi:10.3390/diagnostics10060420_

Round 1

Reviewer 1 Report

This study demonstrated that compared to DSA, VC showed a strong correlation and VFI using VC calculation may be a useful ultrasound method for carotid artery stenosis and stent patency assessment.

This is very interesting study.

My questions are;

How much does angle correction affect?

How much does the skill of sonographers affect the results of VC?

Reviewer 2 Report

Dear authors

Your submission is interesting. however, statistical analysis used in this paper is not appropriate. Pearson correlation, due to the limited sample size and not normally distributed results, can be replaced by Spearman rank correlation.

Best Regards
